# Piezoelectric Gas Sensors with Polycomposite Coatings in Biomedical Application

**DOI:** 10.3390/s22218529

**Published:** 2022-11-05

**Authors:** Anastasiia Shuba, Tatiana Kuchmenko, Ruslan Umarkhanov

**Affiliations:** 1Department of Physical and Analytical Chemistry, Voronezh State University of Engineering Technologies, 394000 Voronezh, Russia; 2Laboratory of Sensors and Determination of Gas-Forming Impurities, Vernadsky Institute of Geochemistry and Analytical Chemistry of Russian Academy of Sciences, 119334 Moscow, Russia

**Keywords:** piezoelectric quartz microbalance, gas sensors, polycomposite coating, mass sensitivity, sorption kinetics, volatile organic compounds, identification, bacterial contamination

## Abstract

When developing methods for diagnosing pathologies and diseases in humans and animals using electronic noses, one of the important trends is the miniaturization of devices, while maintaining significant information for diagnostic purposes. A combination of several sorbents that have unique sorption features of volatile organic compounds (VOCs) on one transducer is a possible option for the miniaturization of sensors for gas analysis. This paper considers the principles of creating polycomposite coatings on the electrodes of piezoelectric quartz resonators, including the choice of sorbents for the formation of sensitive layers, determining the mass and geometry of the formation of sensitive layers in a polycomposite coating, as well as an algorithm for processing the output data of sensors to obtain maximum information about the qualitative and quantitative composition of the gas phase. A comparative analysis of the efficiency and kinetics of VOC vapor sorption by sensors with polycomposite coatings and a set of sensors with relevant single coatings has been carried out. Regression equations have been obtained to predict the molar-specific sensitivity of the microbalance of VOC vapors by a sensor with a polycomposite coating of three sorbents with an error of 5–15% based on the results of the microbalance of VOC vapors on single coatings. A method for creating “visual prints” of sensor signals with polycomposite coatings is shown, with results comparable to those from an array of sensors. The parameters A*_ij_*_∑_ are proposed for obtaining information on the qualitative composition of the gas phase when processing the output data of sensors with polycomposite coatings. A biochemical study of exhaled breath condensate (EBC) samples, a microbiological investigation of calf tracheal washes, and a clinical examination were conducted to assess the presence of bovine respiratory disease (BRD). An analysis of the gas phase over EBC samples with an array of sensors with polycomposite coatings was also carried out. The “visual prints” of the responses of sensors with polycomposite coatings and the results of the identification of VOCs in the gas phase over EBC samples were compared to the results of bacteriological studies of tracheal washes of the studied calves. A connection was found between the parameters A*_ij_*_∑_ of a group of sensors with polycomposite coatings and the biochemical parameters of biosamples. The adequacy of replacing an array of piezoelectric sensors with single coatings by the sensors with polycomposite coatings is shown.

## 1. Introduction

One of the rapidly developing areas of modern sensor technologies is the application of devices based on arrays of diverse sensors in the clinical diagnostics of various pathologies. At the same time, as a rule, the usage of sensors belongs to screening methods in the diagnosis of diseases [1,2]. Among the measuring instruments, a special place is occupied by devices for measuring the gas phase over biosamples—“electronic noses”. Over the past several decades, numerous approaches have been developed for the diagnosis of human and animal diseases by volatile marker analysis [3,4]. These approaches typically involve the detection of specific volatile compounds associated with a specific pathology or the presence of a pathogenic microorganism. These methods often show good specificity and sensitivity, are generally non-invasive, require minimal additional reagents, and can be performed by non-highly qualified personnel. However, there are several problems that must be overcome before such systems can be put into practice, including sampling methodology, reproducibility, and external validation, and the creation and adjunction of a special base of disease-related volatile signatures [1,5,6]. Furthermore, one of the possible problems in the analysis of biological objects may be the limitation of the sample volume, as a result of which the analysis may be impossible or require concentrating. One of the possible solutions to this problem is the development of devices for measuring the gas phase of biological objects without sampling [7,8]. Another option could be the miniaturization of devices based on sensors [9,10], which requires either a fundamental change in the method of measuring and obtaining information from the device, or a reduction in the number of measuring elements. As the dimensions of the device and the number of measuring elements decrease, it is unclear whether it will be possible to maintain the information content of the analysis of biosamples. In the scientific literature, several variants of portable and small-sized devices for the analysis of biological objects with the prospect of diagnosing diseases have been proposed [11,12]. One possible approach to the miniaturization of sensors for the analysis of volatile organic compounds (VOCs) is the creation of polycomposite coatings. Several complex coatings of metal oxide sensors with sensing materials of different structures, such as nanoparticles and nanospheres, have been designed to improve the response, selectivity, or sensitivity of gas analysis [13]. Additionally, composite coatings with metal oxide and polypyrrole particles as chemoresistive gas-sensing materials were used to detect alcohols and organic acids at room temperature [14]. For piezoelectric quartz sensors, a polycomposite coating can be made by a combination of several sorbents that have unique sorption features (kinetic or efficiency) for different VOC classes on one transducer. The development of such coatings is a promising direction in the development of sensor technologies. Since the final characteristics of the sensors are highly dependent on the transducer, the comparative characteristics of various types of sensors are presented in Table 1 [15,16,17,18,19].

Our hypothesis is that it is possible to achieve comparable information content in the analysis of piezoelectric quartz gas sensors with polycomposite coatings by changing the algorithm for processing the output data. Such processing will be sufficient for correct decision making in the diagnosis of diseases.

The goal of this research is to create polymer composite coatings for quartz piezoelectric sensors, which will yield analytical information comparable to that obtained from the existing sensor array. This information will then be used in the diagnosis of diseases.

## 2. Theoretical Background

The method of piezoelectric quartz microbalance is based on the principle of gross weighing of micro quantities of substances, with the possibility of increasing selectivity by choosing modifiers of measuring elements [20]. At the same time, when several sorbents are combined, a synergistic effect is possible in increasing the sensitivity of micro weighing.

Several issues must be solved to create and use polycomposite coatings, including the choice of sorbents for the formation of sensitive layers, the determination of the mass and geometry of sensitive layers from various sorbents in a polycomposite coating, and the development of an algorithm for processing the output data of sensors to obtain maximum information about the quality and quantitative composition of the gas phase. Hereafter, we describe the basic principles for creating such sensors.

### 2.1. Choosing of Sorbents for Polycomposite Coatings

The choice of sorbents for creating a sensitive coating is one of the key factors determining the sorption properties of the sensor. It can be selected empirically, but the most effective way is to choose coatings based on a priori known features of the sorption kinetics of volatile compounds on individual phases. Thus, the choice can be made according to the values of mass sensitivity (*S*_m_, Hz∙m^3^/g) of piezoelectric quartz microbalance of VOC vapors by individual sorbents, since it reflects the change in the sensor signal with an increase in the concentration of a certain substance by 1 g/m^3^, and in fact it is the specific signal of the piezoelectric sensor [21].

It is also important to consider the viscoelastic properties of the resulting thin films when attempting to create a polycomposite coating. The stability of the piezoelectric quartz oscillatory system will be ensured by the closeness of these properties. Because the elasticity coefficients of polymeric and macromolecular sorbents are close, it is possible to create a polycomposite coating using these types of sorbents.

We will demonstrate the suitability of a combination of sorbents for the creation of polycomposite coatings for the detection of volatile organic acids and amines based on the mass sensitivity of these films to diethylamine (DEA) and butanoic acid (BA) vapors. Table 2 shows that sorbent films can be classified into universal ones, which have slightly different affinities for amines and acids (no more than 2–3 times), and selective films, which have an order of magnitude greater affinity for amines or acids.

The sorption affinity for amine and acid vapors can be used to rank universal sorbents, and finer differences in the sorption efficiency of a polycomposite coating based on them can be obtained. The following polycomposite coating can be formed by combining two sorbents (one sorbent/two sorbent, sorbent designation are given in Table 2):One selective + universal sorbents: 18-crown-6/TX-100, 18-crown-6/PEG-2000, 18-crown-6/PEGA;Universal sorbent with greater sensitivity to amines + universal sorbent with greater sensitivity to acids: TX-100/PEGA;Universal with greater sensitivity to acids + selective to amines: PEGA/PDEGS;Universal with greater sensitivity to amines + selective to acids: TX-100/TW;Selective to amines + selective to acids: 18-crown-6/PDEGS;Two universal sorbents: PEG-2000/TX-100.

### 2.2. Choice of the Geometry of the Thin Film Deposition Area and Its Mass

Previously, it was found that an important parameter in the creation of sensors based on piezoelectric quartz resonators is the mass of the applied sorbent [21].

When applying more than two different coatings to the sensor electrode on separate areas of the electrode, the following conditions must be met:The mass of the coating on both sides should be the same or close within the limits of the deposition error (0.1–0.5 µg).The mass of films on separate zones should be the same on each electrode.The total weight of all coatings on both sides must not exceed 20 µg.

If the above conditions are not met, then the mass balance will be violated, and the forces pressing on the quartz plate will shift, lowering the quality factor of the oscillatory system and increasing the noise of the microbalance [24,25].

When creating a sensor with a polycomposite coating, an important point is the ratio of the masses of individual sorbents. Since each sorbent is characterized by certain values of sensitivity and selectivity, by varying the ratios of each of the sorbents, it is then possible to change the sorption properties of the resulting polycomposite coating and sensor based on it. Depending on the properties of the sorbents, the total weight of the polycomposite coating can vary from 10 to 20 µg.

Furthermore, an important parameter for creating polycomposite coatings is the geometry of the area for applying individual sorbents.

There are two main options for where to deposit the sorbent during the formation of a polycomposite coating on the electrode of the measuring element (Figure 1).

The first option is the isolated application of sorbents (Figure 1a), which means that individual sorbents are applied to each side of the electrode of the measuring element. The second option is to alternately apply individual sorbents to each side of the electrode (Figure 1b). In this case, depending on the film formation technique, a new phase may appear at the boundary of two sorbents, which will affect the final sorption properties of the polycomposite coating.

### 2.3. Algorithm for Processing the Output Data of Polycomposite Coatings

The initial characteristics of the sensors, determined by the nature of the sensitive layer of the sensor (electrode coating), to obtain analytical information (qualitative and quantitative indicators) during the sorption of a mixture of vapors of substances on sensors with polycomposite coatings may differ from the output data on the sorption of volatile compounds on an array of sensors with individual films. Therefore, when using sensors with several coatings on one transducer, it is necessary to evaluate and compare the characteristics of the piezoelectric quartz microbalance of vapors of substances by a sensor with a polycomposite coating and a set of sensors with individual films of the same sorbents. It is possible to obtain information about the content of volatile substances in the gas phase above the sample and its composition with a certain probability using the traditional analytical parameters of piezoelectric quartz microbalance (the “visual print” area, S_v.p._, Hz s) and additional sorption parameters (sorption efficiency parameters, A*_i_*_/*j*_).

The sorption efficiency parameter A*_i_*_/*j*_ is one of the sorption parameters that allows for obtaining information about the qualitative composition of a sample [26]. We propose that sorption efficiency parameters can be calculated based on output data from sensors with polycomposite coatings in two ways:According to the signals of the sensor with one polycomposite coating (1/2), as the ratio of its response to 5 s of sorption (∆*F*_(1/2)5_) to the response to 60 s of sorption (∆*F*_(1/2)60_):
A*_ij_*_(5/60)_ = ∆*F*_(1/2)5_/∆*F*_(1/2)60_,(1)

In this case, it is necessary that the sorption kinetics of the substance on the sorbents in the polycomposite coating differ greatly, and the ratio of the mass of sorbents in the coating should be 1:1;
As the ratio of sensor signals with different polycomposite coatings at certain moments of sorption *τ*,s according to the formula:
A*_ij_*_∑_ = (∆*F*_(1/2),*τ*1_/∆*F*_(3/4),*τ*2_): (∆*F*_(5/6),*τ*3_/∆*F*_(7/8),*τ*4_),(2)
where ∆*F*_(1/2),*τ*1_ is the response of the sensor with a polycomposite coating (1/2) at a certain moment of sorption *τ*1.

The metrological characteristics of the method for identifying substances according to the calculated parameters of the sorption efficiency A*_ij_*_(5/60)_, A*_ij_*_∑_ were assessed by calculating the sensitivity and specificity using the formulas [27].
Sensitivity = N_CP_/(N_CP_ + N_FN_),(3)
Specificity = N_CN_/(N_CN_ + N_FP_),(4)
where N_CP_—number of correct positive results, N_FN_—number of false negative results, N_CN_—number of correct negative results, N_FP_—number of false positive results.

## 3. Materials and Methods

### 3.1. Sorbents for Polycomposite Coatings

The sorbents with characteristics presented in Table 2 were selected to create polycomposite coatings. Sensitive zones of sorbents on the electrodes of piezoelectric resonators were formed step-by-step by the drop-casting method described earlier [7].

### 3.2. Volatile Organic Compounds

We used the VOC vapors (organic acids (acetic, butanoic, 2-methylpropanoic, pentanoic, isovaleric), diethylamine, ammonia (25% aqueous solution), and ethanol (puriss, LLC “OS Reachem”, Moscow, Russia)), as well as their model mixtures with different compositions (Table 3) to evaluate the effectiveness of microbalance and study the features of VOC sorption on polycomposite coatings and estimated the influence of the coating formation geometry and the mass of sorbents.

These substances were chosen because they are common volatile markers of pathogenic processes in humans and animals [29,30,31,32,33,34,35,36,37,38,39]. To get closer to the real conditions of analysis, the gas phase was studied not over pure substances, but over their aqueous solutions. Solutions were prepared by diluting the initial substances in bidistilled water, achieving a concentration of 0.01% and 0.1% by volume. Solutions of substances with a volume of 20 cm^3^ were placed in a glass sampler with a volume of 50 cm^3^ with a polypropylene cover and kept for 30 min before analysis at a temperature of 20 ± 2 °C. Gas mixtures were prepared by mixing saturated vapors of the initial substances in 100 cm^3^ flasks with tight polypropylene covers.

### 3.3. Instrument and Measuring Mode

The study of the sorption of compounds on individual (monocoating) and polycomposite coatings of sensors was carried out under static conditions on the device “MAG-8” (OOO “Sensorika—New Technologies”, Voronezh, Russia) [40,41,42], which allows for recording and processing data from 8 sensors in one measurement. The design of the device and the detection cell provide uniform loading of the sensor coatings during measurement. The 3 cm^3^ of the equilibrium gas phase (EGP) was taken with a gas syringe above the solutions of substances and injected into the detection cell of the device. In the special software [36] of the device, the following was recorded: the oscillation frequency of the piezoelectric sensor before the sorption of the substance, the change in the oscillation frequency of the sensor during the sorption of the organic compound with a step of 1 s—a chronofrequency gram, which was used to determine the maximum change in the sensor signal (∆F_max,I_, Hz), and features of the sorption kinetics of substances on the formed polycomposite coatings. The measurement time of the sorption of one sample is 60 s. The detection cell was then purged with dried laboratory air until the initial oscillation frequency of the sensors (the frequency of the sensor oscillation before the measurement began) was reached.

### 3.4. Sensor Output and Processing

The main effectiveness indicators of microbalance include: the maximum change in the sensor signal for a fixed time of substance sorption ∆F_max_, Hz (analytical sensor signal), specific molar sensitivity S_mol_, Hz∙m^3^/mol∙µg, which is calculated by the formula:(5)$$S_{\mathrm{mol}}=\frac{\Delta F_{\max}\cdot M}{c\cdot m_{f}}$$
where *M* is the molar mass of the substance, g/mol; *c* is the concentration of substance vapors in the detection cell, g/m^3^, *m_f_* is the mass of sorbent film.

The possibility of using sensors with individual monocoatings and polycomposite coatings for the identification of vapors of substances in model gas mixtures, and the gas phase over samples was assessed using the sorption efficiency parameter [43,44], which was calculated using the proposed Formulas (1) and (2).

To predict the sorption parameters of sensors with polycomposite coatings, we used the standard multiple regression algorithm [45] realized in MS Excel (LLC “Softline Internet Trade”, Moscow, Russia) by the design of experiment matrices.

### 3.5. The Method of Formation of Polycomposite Coatings and Masses of Sorbents in Them

The design of the experiment was used to determine the geometry of the sorbent deposition area and the mass of each film on the electrodes of the measuring elements, as well as to evaluate the effect of these parameters on the efficiency of VOC sorption.

The design matrix was used to form each of the four polycomposite coatings (1/2)—TW/TX-100, PEG-2000/TX-100, 18-crown-6/PEGA18-crown-6/PDEGS (Table 4).

The analytical signal from sensors with polycomposite coatings was used as a predicted parameter during VOC sorption by the multiple regression method.

### 3.6. Prediction of the Sorption Properties of a Polycomposite Coating

The molar-specific sensitivity S_mol_ was chosen as a predictable factor to forecast the effectiveness of the microbalance of VOC vapors and the sorption properties of polycomposite coatings.

The experiment was carried out in accordance with the full-factor design matrix of experiment 2^3^ for three parameters (sorbents: 18-crown-6, PDEGS, PEGA) and two levels (yes/no), and the mass ratio of sorbents in the coatings was 1:1. The mass of each sorbent in the polycomposite coating was 5.5 ± 2.0 µg with isolated deposition; the mass of sorbents in individual coatings on one sensor was 10.0 ± 1.5 µg. In order to evaluate the suitability of the obtained regression equations, we also studied the sorption of volatile organic compounds on a sensor with an 18-crown-6/PEGA/TX-100 polycomposite coating.

### 3.7. Analysis of Real Objects and Model Gas Mixtures

To test the proposed polycomposite coatings, an array of sensors with two sorbents of different natures (1/2) was selected: PEG-2000/TX-100—sensor 1, TX-100/TW—sensor 2, PEGA/18-crown-6—sensor 3, 18-crown-6/PDEGS—sensor 4. Coated sensors were used in the optimal mass range for each sorbent, considering the geometry of the application area to obtain the maximum analytical signal of the sensor. The choice of these sensors from studied ones with polycomposite coatings was carried out according to the criteria:Minimum system noise during measurement;Operating stability;High sensitivity to volatile disease markers;Possibility of identification of compounds in gas mixtures.

The characteristics of the studied coatings are presented in Table 5.

### 3.8. Collection and Analysis of Biological Samples

The gas phase was studied over samples of exhaled breath condensate (EBC) from calves (*n* = 6) with signs of bovine respiratory disease (BRD) and healthy ones from the respiratory system. The collection EBC samples and animal examination was carried out by a specialist of the All-Russian Scientific Research Veterinary Institute of Pathology, Pharmacology and Therapy (Voronezh, Russia).

All animals were examined according to the clinical scoring system developed by veterinarians at the University of Wisconsin at Madison (Wisconsin respiratory clinical score, WRCS) [46] (measurement of rectal temperature, assessment of the presence of cough, nasal discharge, ocular discharge, and head and ear position). Lung lesions were detected using thoracic auscultation (Littmann^®^ Master Classic II Veterinary Stethoscope, 3 M, Saint Paul, MN, USA). The trachea was palpated [46,47], and a 30 s expiratory apnea was performed to induce coughing in calves [48,49].

In the morning hours before feeding, we received samples of EBC for biochemical studies and gas phase analysis, as well as tracheal washes for bacteriological studies. For the isolation of cultures and typing of microorganeisms, we used meat-peptone broth and agar, milk-salt, enterococcal agar, Endo medium, blood agar, and glucose-serum broth and agar (NICF, St. Petersburg, Russia), in accordance with the established methods. The typing of isolated Escherichia coli was carried out in an agglutination reaction using O-coli sera [50].

Veterinarians collected condensate samples using a special device [51] for collecting exhaled breath into sterile test tubes, froze them in liquid nitrogen, and then delivered them to the laboratory. The test tubes were defrosted at room temperature in containers with tight polystyrene covers for 2 h. The defrosted samples were kept at a temperature of 20 ± 2 °C for 15–30 min before gas phase analysis.

### 3.9. Ethics Statement

The Ethics Committee of the Voronezh State University of Engineering Technologies approved all procedures for clinical examination of the animals and the obtainment of samples for analysis used in this work (Minutes No. 2 dated 25 February 2021). The care and use of animals complied with Russian animal welfare laws, guidelines, and policies; the study did not affect normal animal physiology.

## 4. Results

### 4.1. The Choice of the Method of Formation a Polycomposite Coating and the Mass of Sorbents in Them

Regression equations were obtained for predicting the analytical sensor signals for different geometries and weights of the sorbent film based on the results of vapor-phase organic contaminant sorption on sensors with polycomposite coatings (Table 6).

It has been established that the signal of the sensor with the 18-crown-6/PEGA polycomposite coating is positively affected by an increase in the mass of PEGA. In the sorption of carboxylic acids, the method of coating formation also has a significant effect, and the joint (on one side) deposition of sorbents increases the analytical signal of sorption. Increasing the mass of PDEG in the polycomposite coating reduces the analytical signal of the sensor coated with 18-crown-6/PDEGS during the sorption of organic acids, and increasing the mass of 18-crown-6 and the isolated deposition of sorbents increases the signal of this sensor. For a sensor with a polycomposite coating consisting of TX-100 and PEG-2000, an increase in the mass of PEG-2000 upon the joint deposition of sorbents increases the analytical signal of acid sorption, and an increase in the mass of TX-100 has a positive effect on the analytical signal during the sorption of amines and ethanol. In the case of acid sorption, the analytical signal of a sensor with a polycomposite coating TX-100/TW is positively affected by an increase in the weight of Tween when deposited together onto one side. An increase in the weight of TX-100 increases the analytical signal during the sorption of amines.

Further, the stability of sensors with polycomposite coatings was estimated based on the relative shift in the initial oscillation frequency (*F*_0,0_, MHz) before vapor analysis to the oscillation frequency on *i*-th day of analysis (*F*_0,*i*_, MHz) (Figure 2).

It has been found that the stability of the oscillation frequency of sensors with polycomposite coatings is rather high, and the relative shift in the initial oscillation frequency is no greater than 10–5 relative units for three months of operation. These results are in agreement with the estimation of stability sensors with a single polymer coating, obtained earlier [52].

### 4.2. Evaluation of the Efficiency of VOC Vapor Sorption by a Sensor with a Polycomposite Coating and a Matrix of Relevant Sensors

The results of the microbalance of VOC vapors by sensors with polycomposite coatings and sensors with individual coatings are presented in Table A1 (Appendix B). It should be noted that the repeatability of the sensor signal in the VOC vapor for all studied polycomposite coatings is approximately 15%. To obtain more detailed and objective information about the efficiency of sorption by sensors with polycomposite coatings, the molar-specific sensitivity was calculated. This value provides information about the nature of the interaction between the sorbent film and the adsorbed substance. Figure 3 shows the molar-specific sensitivity of microbalance of vapors of test substances for a polycomposite coating based on a PEGA film.

It has been established that with an increase in the number of sorbents in the coating, an increase in the molar-specific sensitivity of the microbalance of vapors of the test substances occurs, and a sharper increase is typical for the vapors of aliphatic acids. The increase in sensitivity for amines and alcohols is insignificant. The sorption properties of sensors with a polycomposite coating can be estimated using regression equations based on the results of microbalance measurements of volatile organic compound vapors (Table 7).

The error in calculating the molar-specific sensitivity of a polycomposite coating according to the above equations ranges from 5 to 15% for various VOCs.

The resulting predictive equations were tested on the example of a polycomposite coating, in which PDEGS was replaced by triton X-100 (TX-100). According to their chromatographic characteristics, both of these sorbents belong to the group of semipolar ones [53]; therefore, in the first approximation, they can be considered related, although the sensitivity of the microbalance of some substances with thin films of these sorbents differs significantly. The calculated and experimental values of the molar-specific sensitivity of the microbalance of VOC vapors by a sensor with a 18-crown-6/PEGA/TX-100 polycomposite coating are compared in Table 8.

It has been shown that the calculated and experimental values of the molar-specific sensitivity of the microweighing of vapors of the test substances differ from each other by no more than 15%, which indicates good predictive properties of the equations obtained and a correctly chosen quantitative parameter. The kinetic features of sorption are important information for the correct interpretation of the results and can provide additional insight into the efficiency of the microbalance of vapors.

### 4.3. Kinetic Features of Microbalance of VOC Vapors by a Sensor with a Polycomposite Coating and an Array of Relevant Sensors

One of the indicators characterizing the kinetic features of EGP sorption in the static mode is the time to reach the maximum sensor signal (*τ*_max_, s), which must be taken into account when developing an algorithm for recording an analytical signal from a sensor [54,55]. The times to reach the maximum signal of sensors with various individual and polycomposite coatings are presented in Table 9.

It was found that for sensors with polycomposite coatings (1/2), the time to reach the maximum signals corresponds to *τ*_max_ for a sensor with an individual sorbent (Table 9). For a sensor with a polycomposite coating consisting of three sorbents, during the sorption process, several response maxima are observed at different times (Table 9, indicated in parentheses). This indicates a strong contribution of each coating to the response of the sensor with a polycomposite coating as a whole and more complex kinetics interactions with vapors of substances.

This data suggest an algorithm for obtaining analytical information from a sensor with polycomposite coatings from known information for an array of sensors with relevant monocoatings.

Primary analytical information from the sensors can be obtained from the chronofrequency gram—the time dependence of the sensor signals in the process of substance sorption. Figure 4 demonstrates chronofrequency grams of 3-methylbutanoic acid sorption by the sensor with a polycomposite coating and an array of relevant sensors. From Figure 4, it can be seen that the maximum analytical signal from one sensor with monocoating was recorded at the same time, and for each sensor, the signal registration time was different (Table 3, Figure 4b). Therefore, to obtain three analytical signals from an array of sensors, it is necessary to register three signals at selected times. It is possible to obtain the same analytical information from a sensor with a polycomposite coating by recording signals at different time points (5, 40, 60 s from the beginning of sorption—Figure 4a). In this case, the registration time is chosen taking into account the characteristics of the sorption kinetics of a substance. The sorption kinetics on a sensor with a polycomposite coating are close to additive and retain the features of the interaction of individual films with a substance. Moreover, the specificity of the sorption kinetics of VOC at different concentrations is retained, as is shown in Appendix A.

A graphical version of the information from an array of sensors can also be shown as a “visual print” of the sensor signals [40,44]. It is possible to compare the received analytical information from a sensor with a polycomposite coating and an array of relevant sensors in the form of “visual prints” of sensor signals. The form of a “visual print” of responses of a sensor with a polycomposite coating is the same as the form of the “visual print” of maximum sensor signals with a single coating (Figure 5).

However, it is possible to construct “visual prints” based on an a priori assumption about the time of registration of the sensor signal if the polycomposite coating consists of individual sorbents with different sorption kinetics.

In the piezoelectric quartz microbalance method, the amount of a substance in a gas-air mixture can be estimated by the “visual print” area (S_v.p._, Hz s) [40]. It has been established that the content of 3-methylbutanoic acid and ethanol vapors during microweighing with an array of sensors and a sensor with a polycomposite coating coincides within the measurement error.

In real objects, in particular biological samples, substances are contained in trace amounts; the presence of certain gases may indicate the presence of a disease or pathological process [56]. Therefore, in the study of the gas phase over biosamples, a qualitative analysis plays a more important role than a quantitative one. However, there are algorithms for obtaining information about the qualitative composition of mixtures for an array of sensors that need to be optimized when using sensors with a polycomposite coating.

### 4.4. Calculation of Sorption Efficiency Parameters from the Responses of Sensors with Polycomposite Coatings

The calculated sorption efficiency parameters according to Formula (1) for sensors with polycomposite coatings and the sorption efficiency parameters A*_i_*_/*j*_ for sensors with monocoatings are presented in Table 10.

The calculated parameters A*_ij_*_(5/60)_ of a sensor with a polycomposite coating PEGA/PDEGS for VOCs coincide with the calculated parameters A*_i_*_/*j*_ for sensors with individual sorbents within the experimental error.

For a sensor with a polycomposite coating of PEGA/PDEGS, the calculated sorption efficiency parameters for VOCs differ from the A*_i_*_/*j*_ parameters calculated for individual sorbents by no more than 20%. The increased affinity for amines for the TX-100 film explains the significant deviation of the A*_i_*_/*j*_ parameter for triethylamine, calculated from the responses of the sensor with the TX-100/TW polycomposite coating and from the signals of the sensors with the TX-100 and TW films. In this case, it is possible to calculate the sorption efficiency parameters without the possibility of predicting these values from the database of VOC sorption on individual sorbents.

As has been proposed, the sorption efficiency parameters can be calculated using a Formula (2). In this case, the sorbents in the polycomposite coatings of four sensors may be different in nature, and the response time of each sensor with a polycomposite coating may also be different. If the conditions for qualitative criteria are met, the main of which is the independence of the parameter from the substance content in a certain concentration range, the parameter A*_ij_*_∑_ can be considered as identification.

After analyzing the output data from sensors with a polycomposite coating during the sorption of individual VOCs, the identification parameters A*_ij_*_∑_ were chosen (Table 11).

Identification parameters A*_ij_*_∑_ for substances are highlighted in bold in Table 11. For these parameters, all the conditions and assumptions for identification criteria are met [57]; in particular, the values of the parameter are the maximum or minimum values for certain substance, and they differ from the values of this parameter for other substances by at least 3σ. We checked the significance of these parameters when identifying VOCs in model mixtures. Appendix A illustrates the chronofrequency gram of sensors with polycomposite coatings during measurement of model mixtures. The calculated values of the parameters A*_ij_*_∑_ for model mixtures are presented in Table 12.

To assess the error in the identification of substances according to the proposed parameters, the sensitivity and specificity of this approach were calculated, as for systems with a binary response, according to Formulas (3) and (4) (Table 13).

It has been found that the sensitivity of this approach is approximately 70%, and the specificity is 100%. This means that the accuracy of the method is 70%, which is sufficiently high for use in screening methods for diagnosing diseases.

Thus, the algorithm for processing data from sensors with polycomposite coatings can be represented as a scheme (Figure 6).

The resulting algorithm was tested in the analysis of exhaled breath condensate samples to determine respiratory disease in calves.

### 4.5. Analysis of Calf-Exhaled Breath Condensate Samples

The studied calves can be divided into two categories based on the results of the clinical, biochemical, and microbiological methods of analysis: those with BRD (samples 4, 5, 6) and those with a healthy respiratory system (samples 1, 2). Sample 3 refers to a borderline condition, for which there are still no manifested clinical signs of BRD (Table 14). The chronofrequency grams of sensors with polycomposite coatings when measuring the gas phase over breath condensate are presented in Appendix A.

The “visual prints” of the sensor responses for the samples of the calves were constructed to assess the degree of identity in terms of the qualitative and quantitative composition of the samples to each other (Figure 7). To increase the reliability of assessing the state of the respiratory system of the calves according to the data from the sensors with polycomposite coatings, we conducted a qualitative analysis of the composition of EBC with the identification of the main volatile markers of pathogenic microorganisms. The parameters A*_ij_*_∑_ were calculated according to the data from sensors with polycomposite coatings via Formula (3) (Table 15), since these parameters are the most informative for the qualitative composition of the gas phase of the EBC samples.

The parameters of sorption of EGP over EBC samples A*_ij_*_∑_ were compared with biochemical parameters (Figure 8). A negative correlation was found between the concentration of malonic dialdehyde (MDA) in the EBC and the parameter A*_ij_*_∑3_ (R^2^ = 0.9473). It was shown that the concentration of medium molecular weight peptides (MMP) is in a straight-line relationship (R^2^ = 0.9832) with the area of the “visual print” of sensor signals S_v.p_.

## 5. Discussion

An increase in the mass of one or two sorbents in a polycomposite coating generally leads to an increase in the sensor signal, which is consistent with the results of studies of sorption by piezoquartz gas sensors on sorbent films of various masses. However, in some cases, an increase in the mass of the sorbent in a polycomposite coating does not have a positive effect, which depends on the compounds being sorbed and the values of the mass sensitivity of a particular sorbent to them.

When comparing the sorption efficiency of different classes of VOCs, we can conclude that the regularity in sorption for each individual coating is preserved for sensors with polycomposite coatings. This means that the greater the sensor signal with a particular coating for a particular substance, the greater the contribution of that sorbent into the polycomposite-coated sensor signal. Presumably, the increase in S_mol_ with the inclusion of new sorbents in the coating is associated with the synergistic effect of the sorption of substances by two types of sorbents. Thus, when the 18-crown-6 sorbent, which is sensitive to acids, is included in the coating, the molar-specific sensitivity of the microbalance of acid vapors increases sharply (Figure 3).

It has been established that for acids and alcohols, all sorbents are characterized by a significant contribution to the molar-specific sensitivity of the polycomposite coating. For isoacids, a significant contribution is also made by the joint deposition of sorbents (18-crown-6 and PEGA; 18-crown-6 and PDEGS), which indicates a more complex mechanism for the interaction of VOC vapors with a polycomposite coating. A significant contribution of the joint effect of the 18-crown-6 films and PEGA in the polycomposite coating is observed for acetic acid (Table 7). The equations obtained from Table 7 can be applied to calculate the molar-specific sensitivity of a sensor with a polycomposite coating of two sorbents, with an increase in the prediction error of up to 20%. When replacing two or more sorbents in a polycomposite coating, additional modeling is necessary. Thus, according to the S_mol_ values for sensors with monocoatings, it is possible to predict the S_mol_ of a sensor with a polycomposite coating, which is especially important for the identification of organic compounds.

To identify VOCs in the gas phase, the parameters A*_ij_*_(5/60)_, A*_ij_*_∑_ were calculated from the response of sensors with polycomposite coatings. The coincidence of the calculated parameters A*_ij_*_(5/60)_ of a sensor with a polycomposite coating with the parameters A*_i_*_/*j*_ for sensors with individual sorbents indicates that while maintaining the additivity of VOC sorption on polycomposite coatings, it is possible to calculate the sorption efficiency parameters from the responses of a sensor with a polycomposite coating at certain moments of sorption. The calculation of the parameter A*_ij_*_∑_ takes into account the kinetic features of the sorption of substances on a polycomposite coating and in fact is a superposition of two parameters of the sorption efficiency A*_i_*_/*j*_, as selectivity coefficients.

It has been established that all components in mixtures are identified according to the new proposed parameters (Table 12). According to the parameter A*_ij_*_∑1_, ammonia was detected in mixture 3. According to the parameter A*_ij_*_∑3_, triethylamine was identified in mixtures 1, 4, and 5. According to the A*_ij_*_∑5_ parameter, the presence of 2-methylpropanoic acid was established in mixtures No 1–3; in mixtures No. 5 and 7, identification is difficult due to the strong influence of acetic acid on the kinetics. Acetic acid is correctly identified in all mixtures (1, 5–7) by the parameter A*_ij_*_∑7_, which is explained by its significant sorption activity compared to other VOCs. However, for some identification parameters (A*_ij_*_∑2_, A*_ij_*_∑4_, A*_ij_*_∑6_), no test substances were found in the mixtures, which indicates their low sensitivity. It is possible that the limit of detection of these substances is higher than in the studied mixtures for these parameters. The use of these parameters ensures that false-positive identification results are not observed, which confirms the high specificity of their use. Thus, when analyzing biosamples for the presence and content of pathogenic volatile markers, it is possible to replace the array of sensors with a sensor with a polycomposite coating while keeping the information content of analysis.

The primary analytical information obtained by analyzing a sample using an array of sensors is the maximum sensor signals and the “visual prints” built on them. The proportion of various substances in the sample can be estimated by comparing the absolute signals of the sensors. This is because the shape of the “visual print” is determined by the qualitative composition of the volatile fraction of the sample [40].

It has been established that, in terms of the shape of the “visual prints” of the signals of sensors with polycomposite coatings, and, consequently, in terms of the composition of the EGP over EBC, samples 1 and 3 are as close as possible to each other, with a high content of volatile substances in sample 1, which is not associated with a more intensive metabolism during the pathology of the respiratory function (Figure 7). Sample 4 has a different visual print shape than the others, suggesting that the composition of EBC changes without an obvious inflammatory process of the bronchi being diagnosed. This could be a preclinical stage of pneumonia. However, sample 3, which, according to the complex of clinical and laboratory studies, cannot be assigned to a specific diagnostic group (Table 14), is similar in the form of a “visual print” to samples from the group without signs of BRD. Samples 5 and 6 can also be classified as a separate group, which is associated with a change in the composition of the EBC during inflammatory processes in the bronchi and an increase in acidic substances in the EGP above the samples (Figure 7). The features of the geometry of the “visual prints” for samples 5 and 6 may be associated with a population of gram-positive *Enterococcus bacteria* of various species and a few markers of microscopic fungi (*Penicillium* spp., *Asp. Fumigatus*, *Rhizopus nigricans*). At the same time, despite the presence of bacteria in this genus in samples 1 and 4, the shapes of the “visual prints” for these samples differ due to the presence of other microorganisms (*Staph. Epidermidis*, *E. coli*). To confirm this hypothesis, VOCs were identified in the gas phase over the EBC samples using the A*_ij_*_∑_ parameters.

According to the previously established values of identification parameters A*_ij_*_∑_ for VOCs, the presence of acetic acid in all samples was determined by parameter A*_ij_*_∑7_ (0.8), since it is one of the volatile markers of conditionally pathogenic microorganisms, such as *Staphylococcus* ssp., *E. Coli* [6], found in the tracheal washes of the studied calves (Table 14 and Table 15). In samples 3 and 4, triethylamine (2.0), which is a volatile marker of infection [58], for example, *Enterococcus* ssp., was found by parameter A*_ij_*_∑3_. Previously, it was identified by other parameters of sorption efficiency A*_ij_*_∑_ in all samples, since bacteria of this genus, according to bacterial studies, are present in the washes of the studied calves. In addition, in samples 3 and 4, ammonia was identified by the parameter A*_ij_*_∑1_ (2.0), which is a sign of the initial stage of the inflammatory process of tissues during the hydrolysis of their protein structure [59]. The 2-methylpropanoic acid was identified in samples 5 and 6 by the parameter A*_ij_*_∑5_ (0.4), which is a gas marker of gram-negative bacteria and microscopic fungi [60].

The presence of isocompounds in the EGP above the biosamples indicates the occurrence of a pathogenic process [61], in this case, acute or chronic bronchopneumonia. When developing new methods for analyzing real objects using new or modified measuring systems, it is necessary to assess the adequacy of the results obtained with regard to traditional methods of analysis. An increase in MDA in exhaled breath condensate indicates the development of inflammation in the airways, reaching a maximum value in the chronic stage of the disease [62]. Therefore, using the identification parameter for triethylamine, it is possible to estimate the content of MDA in EBC (Figure 8). An important biochemical indicator of condensate in the diagnosis of pneumonia is the concentration of medium-molecular weight peptides (MMP) [63], which can be estimated from the area of the “visual print” of sensor signals (Figure 8). The obtained information on the qualitative and quantitative composition of the equilibrium gas phase above the EBC samples is comparable to the analytical information obtained from an array of eight piezosensors [64], which is necessary for a correct assessment of the presence of BRD in calves.

Thus, it is possible to use an array of sensors with polycomposite coatings for diagnosing an inflammatory process in the respiratory organs of calves by the composition of the equilibrium gas phase over samples of exhaled breath condensate, while maintaining the same information content as with an array of eight piezosensors.

## 6. Conclusions

In this work, a comparative analysis of the efficiency and kinetics of VOC vapor sorption by sensors with polycomposite coatings and a matrix of relevant sensors was carried out in terms of the main microbalance parameters: molar specific sensitivity and the time to reach the maximum sensor signal. Regression equations have been obtained to predict the molar specific sensitivity of the microbalance of VOC vapors by a sensor with a polycomposite coating of three sorbents with an error of 5–15% based on the results of the microbalance of VOC vapors on monocoatings.

A method for constructing “visual prints” of sensor signals with polycomposite coatings in the analysis of model mixtures is shown. Its adequacy and applicability in the analysis are shown. Parameters A*_ij_*_∑_ are proposed to obtain information on the qualitative composition of the gas phase when processing the output data of sensors with polycomposite coatings. The sensitivity and specificity of this approach are 70% and 100%, respectively. Based on research into the comparative characteristics of sensor arrays and sensors with polycomposite coatings, the results are presented in Table 16.

A study of the equilibrium gas phase over samples of exhaled breath condensate of calves was carried out to assess the presence of BRD. “Visual prints” of signals from sensors with polycomposite coatings for the samples of exhaled breath condensate were constructed, and a qualitative analysis of the EGP over biosamples was carried out according to the identification parameters A*_ij_*_∑_. The data obtained were analyzed taking into account bacteriological studies of tracheal washes of the studied calves for the presence of microorganisms.

The adequacy, information content, and the possibility of replacing an array of piezosensors with monocoatings with an array of sensors with polycomposite coatings for diagnosing an inflammatory process in the respiratory organs of calves by the composition of the equilibrium gas phase over samples of exhaled breath condensate are shown. A correlation of sorption parameters from an array of sensors with polycomposite coatings with biochemical parameters of biosamples has been established.

## Figures and Tables

**Figure 1 sensors-22-08529-f001:**
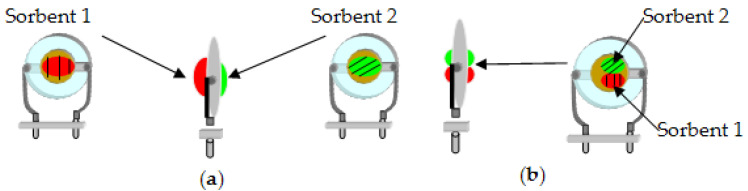
Technique of the formation of polycomposite coatings: isolated deposition—(**a**), joint deposition—(**b**).

**Figure 2 sensors-22-08529-f002:**
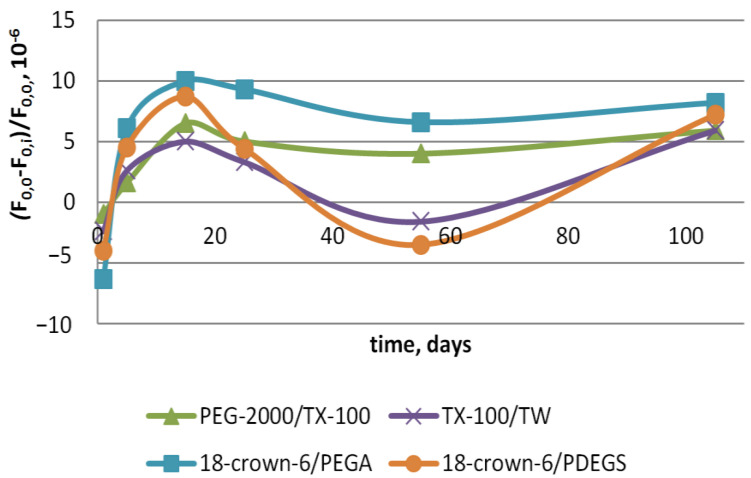
Dependence of relative shift of initial oscillation frequency ((*F*_0,0_ − *F*_0,*i*_)/*F*_0,0_) of sensor with polycomposite coatings on day of analysis.

**Figure 3 sensors-22-08529-f003:**
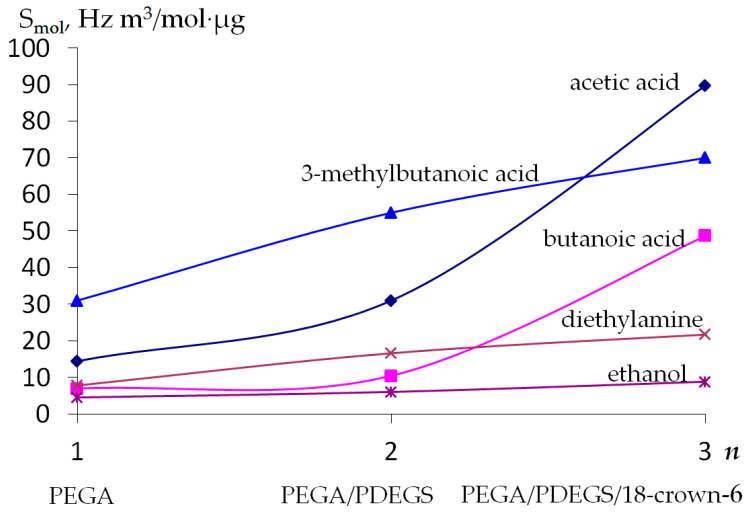
Dependence of the molar-specific sensitivity (S_mol_) of microbalances of vapors of various substances on the number of sorbents (*n*) in a polycomposite coating.

**Figure 4 sensors-22-08529-f004:**
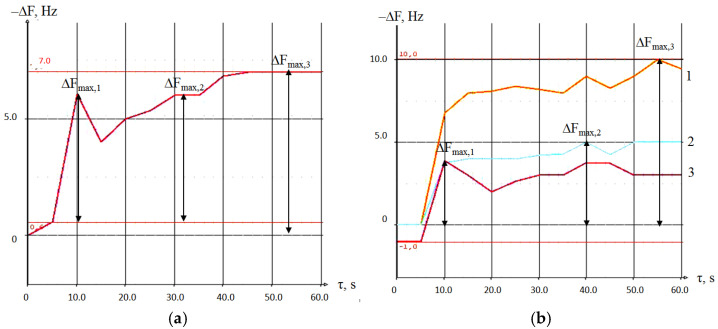
Chronofrequency grams of 3-methylbutanoic acid vapor sorption by a sensor with a polycomposite coating 18-crown-6/PEGA/PDEGS (**a**) and an array of sensors (**b**) with films 18-crown-6 (1), PEGA (2), PDEGS (3).

**Figure 5 sensors-22-08529-f005:**
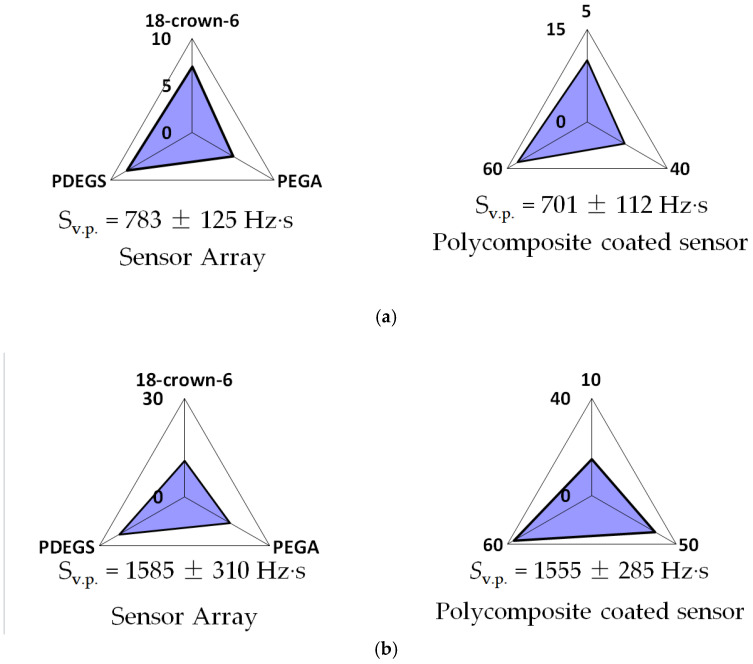
“Visual prints” by maximum signals for an array of sensors with individual coatings and considering the sorption kinetics for a sensor with a polycomposite coating during microweighing of vapors of 3-methylbutanoic acid (**a**) and ethanol (**b**). For an array of sensors, sorbents are marked in a circle; for a sensor with a polycomposite coating, the time of signal registration is marked.

**Figure 6 sensors-22-08529-f006:**
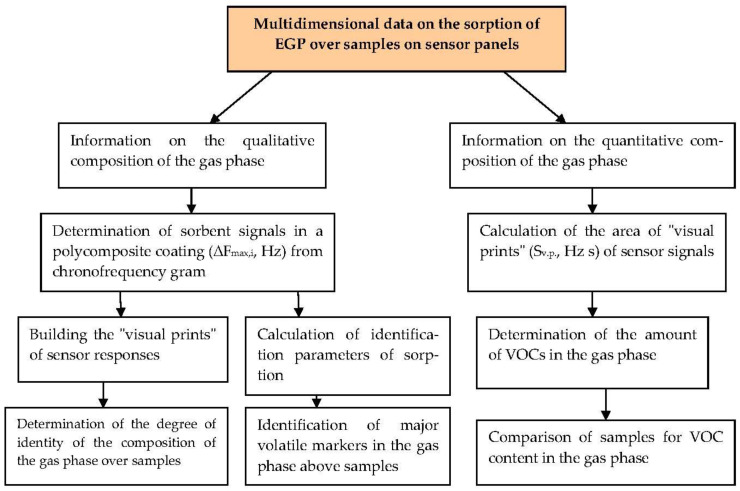
Scheme of the algorithm for processing output data from sensors with polycomposite coatings.

**Figure 7 sensors-22-08529-f007:**
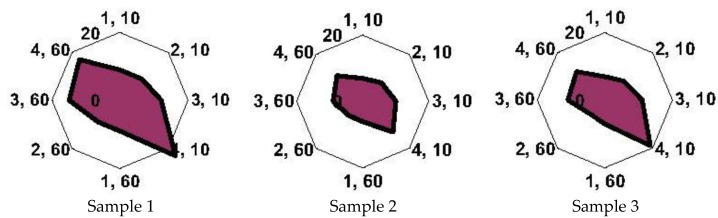
“Visual prints” of sensor responses in an array with polycomposite coatings in the EGP over EBC samples: numbers of sensors with polycomposite coatings in an array and scanning time (10, 60 s) are plotted along the radial axis. The part of “visual print” reflected the presence of disease in calf is selected by blue circles.

**Figure 8 sensors-22-08529-f008:**
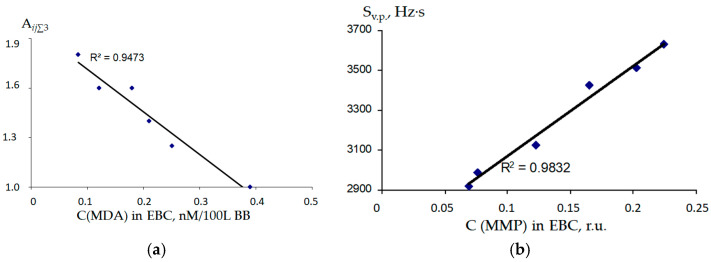
Dependencies of the sorption parameters of an array of sensors with polycomposite coatings on the biochemical parameters of the EBC: (**a**) the A*_ij_*_∑3_ parameter on the MDA content; (**b**) the area of the “visual print” (S_v.p._) on the content of the MMP.

**Table 1 sensors-22-08529-t001:** Comparative characteristics of various types of sensors.

Type of Sensor	Limit of VOC Detection	Advantages	Disadvantages
Chemoresistive	5–500 ppm	High sensitivity, low operating temperature, and a thermal stable structure, simplicity, low cost, small size and ability to be integrated into electronic devices	High sensitivity to water vapor, high possibility of sensor poisoning, low selectivity
Optical	1 ppm–1000 ppb	Commercial availability, simplicity of sensor formation	The complexity of creating devices, fluorescent dyes have a short operating time
Metal oxide	1–1000 ppm	Low power consumption, the possibility of long battery life, long life of the sensor material, ability to work in explosive environments	Low selectivity, poor sensitivity to organic molecules and relatively low stability caused by recrystallization and surface poisoning processes
Piezoelectric quartz microbalance	10 ppm–10 ppb	Linear calibration curve over a wide concentration range, fast response and recovery time, high sensitivity	Fragile sensing element, possibility of electrode corrosion
Surface acoustic waves (SAW)	1 ppm–1 ppb	High sensitivity, excellent response time, small size, low cost, ability to work in wired and wireless mode	Membrane aging

**Table 2 sensors-22-08529-t002:** Characteristics of sorbents for modifiers of electrodes of piezoelectric quartz resonators.

Name	Producer	Designation	Solvent	Type of Sorbent	*S*_m_ *, Hz∙m^3^/g (for BA)	*S*_m_ *, Hz∙m^3^/g (for DEA)
Polyethylene glycol 2000	Alfa Aesar, Ward Hill, MA, USA, p.a.	PEG-2000	Acetone	Universal	28.8	25.5
Triton X-100	Alfa Aesar, Ward Hill, MA, USA, p.a.	TX-100	Acetone		27.5	43.7
Polyethylene glycol adipate	Reachem, Moscow Russia, (puriss.)	PEGA	Acetone		41.3	19.4
Dicyclohexyl-18-crown-6	Alfa Aesar, Ward Hill, MA, USA, p.a.	18-crown-6	Toluene	Selective to acid	77.5	4.7
Polyoxyethylene Sorbitan Monopalmitate, Tween 40	Reachem, Moscow Russia, (puriss.)	TW	Toluene		71.3	5.1
Polydiethylene glycol succinate	Reachem, Moscow Russia, (puriss.)	PDEGS	Acetone	Selective to amines	8.3	98

* estimated experimentally early [22,23].

**Table 3 sensors-22-08529-t003:** Composition of model gas mixtures of volatile organic compounds.

Mixture Number	Ammonia (0.775 * g/m^3^)	Triethylamine (0.0014 g/m^3^)	Acetic Acid (0.0013 g/m^3^)	2-methylpropanoic Acid (0.0006 g/m^3^)
1	+	+	+	+
2	+	−	−	+
3	+	+	−	+
4	+	+	−	−
5	−	+	+	+
6	−	+	+	−
7	+	−	+	+
8	−	−	−	−

*—concentration is calculated by Antoine equation [28].

**Table 4 sensors-22-08529-t004:** Design matrix for the choice of masses of sorbents and the technique of forming a polycomposite coating.

No.	Masses of Sorbents in a Coating		Technique of Formation (TF)
	m_1_ ± 0.15, µg	m_2_ ± 0.15, µg	(“+”—Isolated, ”−”—Joint)
1	10	10	+
2	5	10	−
3	10	5	−
4	5	5	+

**Table 5 sensors-22-08529-t005:** The composition of the polycomposite coating for the used sensors.

Polycomposite Coating (1/2)	Mass of Sorbent 1 in Coating, µg	Mass of Sorbent 2 in Coating, µg	Technique of Formation
18-crown-6/PEGA	4.7	7.5	joint deposition
TX-100/TW	5.3	11.3	joint deposition
18-crown-6/PDEGS	6.9	3.6	isolated deposition
PEG-2000/TX-100	7.5	1.9	joint deposition

**Table 6 sensors-22-08529-t006:** Regression equations for predicting the analytical signals of sensors (∆F_max_, Hz) based on polycomposite coatings depending on the mass of sorbents (m_1_, m_2_) and technique of formation (TF) (*p* = 0.95, *n* = 3).

VOC	Polycomposite Coating (1/2)			
	18-crown-6/PEGA	18-crown-6/PDEGS	TX-100/PEG-2000	TX100/TW
Ethanoic acid	∆F_max_ = 14 + 2 m_2_ − 4 TF	∆F_max_ = 10 − 2.5 m_2_	∆F_max_ = 21 + 3 m_2_ − 4 TF	∆F_max_ = 10 + 2 m_2_
Butanoic acid	∆F_max_ = 12+ 2 m_2_ − 5 TF	∆F_max_ = 9.5 + 3 m_1_ − 2.5 m_2_	∆F_max_ = 19 + 5 m_2_ − 3 TF	∆F_max_ = 9 + 1.5 m_2_
2-methylpropanoic acid	- *	-	∆F_max_ = 24 + 3 m_2_ − 3 TF	∆F_max_ = 11.5 + 1.5 m_2_
Pentanoic acid	∆F_max_ = 14 + 3.5 m_2_ − 3 TF	∆F_max_ = 11 − 3 m_2_ + TF	-	-
3-methylbutanoic acid	∆F_max_ = 14 + 2 m_2_ − 4 TF	∆F_max_ = 11 + 2 m_1_ − 4 m_2_	∆F_max_ = 20 + 3 m_2_ − 7 TF	∆F_max_ = 8 + m_2_ − 2 TF
Ammonia	-	-	∆F_max_ = 30 − 8m_1_ − 7 m_2_	∆F_max_ = 11 + 2 m_1_ + 3 m_2_
Diethylamine	-	∆F_max_ = 17.5 − 4.5 m_2_	∆F_max_ = 46 − 7m_1_ + 12 TF	∆F_max_ = 14 + 3 m_1_ + 4 m_2_
Ethanol	∆F_max_ = 21 + 2 m_1_ + 6 m_2_	∆F_max_ = 16.5 − 4.5 m_2_ + 3.5 TF	∆F_max_ = 33 + 5.5 m_1_ − 3 TF	∆F_max_ = 21 + 3 m_1_ + 4.5 m_2_

*—it is impossible to make an equation at this level of significance.

**Table 7 sensors-22-08529-t007:** Regression equations for predicting the molar-specific sensitivity (S_mol_) of microbalance of VOC vapors by a sensor with a polycomposite coating 18-crown-6/PEGA/PDEGS.

VOC	Equation
Acetic acid	$S_{mol}^{}$ = 47 + 0.27 $S_{mol}^{}$ _(18-crown-6)_ + 0.09 $S_{mol}^{}$ _(PDEGS)_ − 0.003 $S_{mol}^{}$ _(18-crown-6)_ $S_{mol}^{}$ _(PEGA)_
Butanoic acid	$S_{mol}^{}$ = 2.0 + 1.3 $S_{mol}^{}$ _(18-crown-6)_ + 0.65 $S_{mol}^{}$ _(PDEGS)_ + 1.1 $S_{mol}^{}$ _(PEGA)_
3-methylbutanoic acid	$S_{mol}^{}$ = 48 + 0.13 $S_{mol}^{}$ _(18-crown-6)_ + 0.14 $S_{mol}^{}$ _(PDEGS)_ − 0.0006 $S_{mol}^{}$ _(18-crown-6)_ $S_{mol}^{}$ _(PEGA)_ − 0.0005 $S_{mol}^{}$ _(18-crown-6)_ $S_{mol}^{}$ _(PDEGS)_
Diethylamine	$S_{mol}^{}$ = 1.3 + 0.12 $S_{mol}^{}$ _(18-crown-6)_ + 0.64 $S_{mol}^{}$ _(PDEGS)_
Ethanol	$S_{mol}^{}$ = 5.6 + 0.07 $S_{mol}^{}$ _(18-crown-6)_ + 0.17 $S_{mol}^{}$ _(PDEGS)_ + 0.04 $S_{mol}^{}$ _(PEGA)_

**Table 8 sensors-22-08529-t008:** Calculated and experimental molar-specific sensitivity (S_mol_) of microweighing of VOC vapors by a sensor with polycomposite coating 18-crown-6/PEGA/TX-100.

Substance	$S_{mol}^{}$ Experimental	$S_{mol}^{}$ Calculated	Error, %
Acetic acid	104	96.8	6.70
Butanoic acid	41.8	47.3	13.2
3-methylbutanoic acid	101	88.6	12.1
Diethylamine	15.3	14.1	8.02
Ethanol	8.91	7.92	11.1

**Table 9 sensors-22-08529-t009:** Time to reach the maximum signal (*τ*_max_, s) with individual coatings and polycomposite coatings upon sorption of vapors of VOCs.

VOC	Individual Coating			Polycomposite Coating (1/2/3)			
	18-crown-6	PEGA	PDEGS	18-crown-6 /PEGA	PEGA/ PDEGS	18-crown-6/ PDEGS	18-crown-6/PEGA/ PDEGS
Acetic acid	60	60	60	60	60	60	60
Butanoic acid	40	60	5	30	60	15	10 (60) *
3-methylbutanoic acid	60	40	5	60	40	5	5 (60)
Diethylamine	60	30	60	60	60	60	30 (60)
Ethanol	60	10	60	60	60	60	10 (60)
Water	30	15	60	10	60	60	60

*—additional time for the maximum sensor signal is indicated in brackets.

**Table 10 sensors-22-08529-t010:** VOC sorption efficiency parameters A*_ij_*_(5/60)_ for polycomposite coatings and an array of relevant sensors (*n* = 3, S_r_(0.05) ≤ 0.2).

VOC	Signals Sensors with Individual Sorbent 1 and 2		A_1/2_	Response of Sensor with Polycomposite Coating (1/2) for 5 and 60 s of Sorption		A_12(5/60)_	∆, %
	∆*F*_1_ ± 2, Hz	∆*F*_2_ ± 2, Hz		∆*F*_(1/2),5_	∆*F*_(1/2),60_		
	1—PEGA	2—PDEGS		PEGA/PDEGS			
Acetic acid	8	14	0.6	12	22	0.5	20
2-methylpropanoic acid	9	13	0.7	7	10	0.7	0
Ammonia (1% vol. aqua solution)	11	60	0.2	9	50	0.2	0
Triethylamine	84	92	0.9	65	82	0.8	13
Ethanol	18	25	0.7	22	35	0.6	17
Butanol	12	20	0.6	17	25	0.7	14
	1—Tween	2—TX-100		TX-100/Tween			
Acetic acid	6	10	0.6	8	13	0.6	0
2-methylpropanoic acid	7	9	0.8	6	12	0.5	60
Ammonia (1% vol. aqua solution)	13	7	1.9	14	7	2.0	5
Triethylamine	35	10	3.5	19	10	1.9	84
Ethanol	10	17	0.6	15	23	0.7	14
Butanol	8	15	0.5	7	15	0.5	0

**Table 11 sensors-22-08529-t011:** Parameters A*_ij_*_∑_ for VOCs calculated considering the kinetic features of sorption on sensors with polycomposite coatings (*n* = 3, S_r_(0.05) < 0.15).

Designation	Equation A*_ij_*_∑_	Acetic Acid	2-Methylpropanoic Acid	Triethylamine	Ammonia
A*_ij_*_∑1_	(∆*F*_1,10_ */∆*F*_3,40_): (∆*F*_2,16_/∆*F*_4,16_)	0.6	0.8	1.5	**2.0**
A*_ij_*_∑2_	(∆*F*_2,12_/∆*F*_3,40_): (∆*F*_3,40_/∆*F*_4,40_)	0.3	2.0	0.5	**6.7**
A*_ij_*_∑3_	(∆*F*_1,10_/∆*F*_3,12_): (∆*F*_2,16_/∆*F*_4,16_)	1.0	0.7	**2.0**	0.9
A*_ij_*_∑4_	(∆*F*_2,40_/∆*F*_4,10_): (∆*F*_2,10_/∆*F*_3,40_)	**2.3**	0.8	1.6	0.4
A*_ij_*_∑5_	(∆*F*_1,16_/∆*F*_2,10_): (∆*F*_2,10_/∆*F*_3,40_)	**5.0**	**0.4**	2.0	1.5
A*_ij_*_∑6_	(∆*F*_2,12_/∆*F*_4,12_): (∆*F*_2,16_/∆*F*_4,16_)	1.0	**1.8**	0.7	0.4
A*_ij_*_∑7_	(∆*F*_2,16_/∆*F*_4,16_): (∆*F*_2,40_/∆*F*_4,40_)	**0.8**	1.2	1.8	1.2

∆*F*_1,10_ *—response of sensor 1 described in p. 2.7, time of registration—10 s. The values of parameters for identification of substances are highlighted in bold.

**Table 12 sensors-22-08529-t012:** Parameters A*_ij_*_∑_ for model mixtures, calculated considering the kinetic features of sorption on sensors with polycomposite coatings (*n* = 3, Sr (0.05) < 0.15).

Designation	Number of Mixture								Coincidence Criterion *d*
	1	2	3	4	5	6	7	8	
A*_ij_*_∑1_	*2.2*	*2.0*	1.6	*1.8*	1.6	1.2	*2.2*	1.3	0.2
A*_ij_*_∑2_	4.0	3.5	3.8	2.8	0.9	0.8	2.8	1.0	0.6
A*_ij_*_∑3_	*2.4*	3.0	1.7	*2.3*	*2.4*	1.3	3.4	1.5	0.4
A*_ij_*_∑4_	0.4	0.4	0.5	0.5	0.7	0.7	0.5	0.6	0.3
A*_ij_*_∑5_	*0.5*	*0.6*	*0.4*	0.7	1.4	1.1	0.7	0.9	0.2
A*_ij_*_∑6_	1.2	1.0	1.1	1.0	1.0	0.8	1.2	1.0	0.3
A*_ij_*_∑7_	*1.0*	1.2	1.2	1.2	*1.0*	*1.0*	*1.0*	1.2	0.2

Note: italicized values of the parameters match the identification ones within the coincidence criterion *d*.

**Table 13 sensors-22-08529-t013:** Sensitivity and specificity of VOC identification by A*_ij_*_∑_ parameters in model mixtures.

Substance	Parameter A*_ij_*_∑_	Sensitivity, %	Specificity, %
Ammonia	A*_ij_*_∑1_	73	100
Triethylamine	A*_ij_*_∑3_	67	100
Acetic acid	A*_ij_*_∑5_	100	100
2-methylpropanoic acid	A*_ij_*_∑7_	73	100

**Table 14 sensors-22-08529-t014:** Results of clinical examination and biochemical analysis of exhaled breath condensate samples of calves.

Indicator		Number of Calf/Biosample					
		1	2	3	4	5	6
WRSC		2	3	3	5	8	7
Presence BRD		−	−	+/−	+	+	+
Medium molecular weight peptides in EBC, r.u.		0.076	0.203	0.069	0.123	0.165	0.224
Malonic dialdehyde in EBC, nM/100 L BB		0.121	0.210	0.084	0.180	0.251	0.390
pH EBC		7.35	7.34	7.32	7.58	7.63	7.66
Presence in a tracheal wash	*Ent. faecium*	poor growth	poor growth	Moderate growth	-	-	-
	*Staph. Epidermidis*		-	-	-	Moderate growth	
	*Ent. faecalis*	-	-	Moderate growth			
	*E. coli*	-	-	strain O115 Moderate growth		O9, O26 Moderate growth	O115, O8 Moderate growth
	*Penicillium* spp.	-	-	-	-	poor growth	-
	*Asp. fumigatus*	-	-	-	-	-	slightly.
	*Rhizopus nigricans*	-	-	-	-	-	

**Table 15 sensors-22-08529-t015:** Parameters A*_ij_*_∑_ for EBC samples, calculated considering the kinetic features of sorption on sensors with polycomposite coatings (*n* = 4–5, S_r_(0.05) < 0.15).

Designation	Sample Number						*d*
	1	2	3	4	5	6	
A*_ij_*_∑1_	1.2	1.4	*1.8*	*1.7*	1.2	1.0	0.2
A*_ij_*_∑2_	0.6	1.0	0.7	0.6	1.0	0.8	0.2
A*_ij_*_∑3_	1.5	1.4	*1.8*	*2.0*	1.2	1.0	0.4
A*_ij_*_∑4_	0.7	0.5	0.4	1.0	0.4	0.4	0.3
A*_ij_*_∑5_	1.3	0.8	1.0	1.6	*0.6*	*0.6*	0.2
A*_ij_*_∑6_	0.6	1.0	1.0	1.3	1.0	1.0	0.3
A*_ij_*_∑7_	*0.8*	*1.0*	*0.8*	*0.6*	*1.0*	*0.8*	0.2

Note: italicized values of the parameters match the identification ones within the criterion *d*.

**Table 16 sensors-22-08529-t016:** Comparison of using sensors array and polycomposite coatings.

Characteristics	Sensor Array	Polycomposite Coatings
Specific molar sensitivity, S_mol_, Hz m^3^/mol∙μg	40–1500	40–2500
Selectivity	Depends on sensitivity of sorbents	Depends on sensitivity of sorbents
Response time	Until 60 s	Depends on kinetic of sorption of selected sorbents
Time of full regeneration of sensors surface into closed detection cell	2–3 min	1 min
Keeping the analytical information at miniaturization of detection cell in 2–10 times	No	Yes
Number of studied sorbents per one measurement	8	16–24

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
