# Peer review of "Piezoelectric Gas Sensors with Polycomposite Coatings in Biomedical Application"

_sensors, 2022, doi:10.3390/s22218529_

Round 1

Reviewer 1 Report

This manuscript reported a polycomposite coated piezoelectric gas sensor for biochemical application. The background is clear and the experiments are designed appropriately. But there are still some issues to be well addressed. I hope the following comments may help.

1.     Please carefully check the arrangement of figures, for example, Figure 1 is blank (cannot shown) in the PDF.

2.     I am wondering the specific sensing mechanism of the polycomposite toward the detected VOCs, please clarify this.

3.     How are the sensing repeatability and stability after cycle test and aging test? Besides, more experimental data should be provided such as the performance under different gas concentrations.

4.     As claimed in the introduction, one important trend is the miniaturization of the devices, so in this work polycomposite coatings are created, could the author provide any evidence of the sensing materials and the device to support the view of miniaturation?

5.     Compared with the previous reports, the novelty or improvements on the sensing performance should be clarified, a summarized Table is sugguested.

Author Response

The authors are grateful for the study of the manuscript and valuable comments. Regarding the questions the following explanations are made:

Point 1. Please carefully check the arrangement of figures, for example, Figure 1 is blank (cannot shown) in the PDF.

Response 1: The manuscript has been corrected, figure 1 has been included in the text.

Point 2. I am wondering the specific sensing mechanism of the polycomposite toward the detected VOCs, please clarify this.

Response 2: The mechanism of detection of volatile compounds by polycomposite coatings is as follows:

  1. The features of sorption of volatile compounds on individual coatings (one film per sensor) are studied: the time to reach the maximum signal, mechanism of sorption - surface (adsorption mechanism) or accumulation (absorption mechanism).
  2. A polycomposite coating is formed from the studied sorbents. Since the substance reacts simultaneously with all sensors in the detection cell, the specificity of the detection of volatile compounds is achieved through an processing algorithm taking into account the characteristics of the sorption of substances, i.e. the polycomposite coating does not specifically interact with the substance under study, however, after a detailed study of the mechanism of sorption on individual coatings, this information can be used to assess the presence of VOCs in gas mixtures or gas phases over real objects.

Our hypothesis is that when applying several sorbents to one transducer and reading its responses during the sorption of substances, the features of the sorption kinetics of this substance are preserved and they can be distinguished from the chronofrequencyogram by reading the sensor signals at certain moments of sorption.

Point 3. How are the sensing repeatability and stability after cycle test and aging test? Besides, more experimental data should be provided such as the performance under different gas concentrations.

Response 3: As an estimate of the reproducibility of sensor signals when measuring VOC, the value of the confidence interval was used, which is added to Table 1a. Studies on polycomposite coatings were carried out for three months, during this time no significant deviations in the sensor signals associated with film aging were found. In addition, we studied these coatings as sorbents earlier, and it was shown that the drift of the baseline of sensors over half a year in the analysis of VOCs with a low concentration is no more than 10%. The results of evaluating the operating time of polymer coatings are presented in the article T. A. Kuchmenko, A. A. Shuba, and E. V. Drozdova // Substantiation of the Operating Life of Gas Piezosensors in Detection of Vapors of Organic Compounds, Russian Journal of Applied Chemistry, vol. 88, pp. 1997-2008 Dec. 2015, doi: 10.1134/S10704272150120150. In this article, sorbents are studied in terms of structural and viscoelastic properties similar to those proposed in the manuscript. The change in the baseline of sensors with polycomposite coatings and a reference to the article were added to the manuscript in the Results section.

Since it is proposed to use sensors with polycomposite coatings for the analysis of biological objects in which volatile compounds are contained in relatively low concentrations, we studied the sorption of VOCs at low concentrations (solutions of substances with concentrations from 1% to 0.01% by volume). Chronofrequency grams for substances at various concentrations are presented in Supplement materials, Figure 1S. A reference to the figure has been added to the Results section.

Point 4. As claimed in the introduction, one important trend is the miniaturization of the devices, so in this work polycomposite coatings are created, could the author provide any evidence of the sensing materials and the device to support the view of miniaturation?

Response 4: This manuscript presents the results on the sorption of substances in a detection cell with a volume of 154 cm3, similar trends were established when using sensors with polycomposite coatings in a detection cell with a volume of 64 cm3. Nowadays, we are going to carry out the experiment with detection cell of 10cm3, but unfortunately, we are not be able to provide these results. We suggest both the possibility of miniaturization of devices and the use of polycomposite sensors in usual cells detection, with an increase in information content due to the use of a larger number of sorbents on 8 transducers. Briefly, these results are added in general form in table No. 16 in the Conclusion section.

Point 5. Compared with the previous reports, the novelty or improvements on the sensing performance should be clarified, a summarized Table is sugguested.

Response 5: Comparison parameters and advantages of using polycomposite sorbents in gas analysis are given in Table 16 in the Conclusion section.

Reviewer 2 Report

In this paper, a new method of VOC detection by vapor adsorption is reported, and the samples of exhaled gas from animals are tested. At the same time, the prediction regression equation of molar ratio sensitivity between VOC and composite coating sensor of three adsorbents is established. The sensor works well and has medical practical value. Here are some questions:

1. The digital image of Figure 1 is missing.

2. In line 123, the authors refer to the combination of two generic adsorbents. How is this different from the two generic sorbent combinations mentioned on line 119?

3. In lines 209-217, the authors refer to diluting the initial substance in double distilled water.  Please indicate whether the presence of water vapor affects VOC detection.

4. Author proposed that polycomposite coating was used to detect VOC and carried out a control experiment with a single coating, but did not specifically point out the advantages of polycomposite coating over a single coating.

5. The article mentioned collecting the exhaled gas of animals for sample analysis. Please explain how to selectively detect the required VOC among the many exhaled gas molecules. Please address the selectivity of the sensors.

6. Please explain the detection mechanism of the sensor.

7. Some recent works related to VOC sensors are missing in the references. Coordination Chemistry Reviews 452 (2022) 214280; Sensors and Actuators A: Physical 2022, 347,113933. And it would be better to comment on the main advantages/disadvantages of the different sensor types.

Author Response

The authors are grateful for the study of the manuscript and valuable comments. Regarding the questions the following explanations are made:

Point 1. The digital image of Figure 1 is missing.

Response 1: The manuscript has been corrected, figure 1 has been included in the text.

Point 2. In line 123, the authors refer to the combination of two generic adsorbents. How is this different from the two generic sorbent combinations mentioned on line 119?

Response 2: There is a slight difference between the mentioned coatings, in the case of TX-100/PEGA - sorbents with a small difference in mass sensitivity to amines and acids are used, in the case of PEG-2000/TX-100 - one of the sorbents (PEG-2000) has no difference in mass sensitivity to acids and amines. Therefore, we can say that there is no fundamental difference in the approach to the formation of these coatings, only the possibility of slightly changing the sensitivity to the classes of compounds.

Point 3. In lines 209-217, the authors refer to diluting the initial substance in double distilled water.  Please indicate whether the presence of water vapor affects VOC detection.

Response 3: Of course, the presence of water vapor is an interfering factor in the determination of volatile compounds using an array of sensors. However, the mass sensitivity of films of selected sorbents to water vapor is less than to vapors of other VOCs. Also, the effect of water vapor on the possibility of detecting VOC vapors from the output data of sensors with the proposed films was considered earlier in the articles:

  1. Selection of a piezoelectric sensor array for detecting volatile organic substances in water / A. A. Shuba, T. A. Kuchmenko, E. I. Samoilova, N. V. Bel'skikh // Moscow University Chemistry Bulletin. - 2016. - Vol. 71. - No 1. - P. 68-75. – DOI 10.3103/S0027131416010156. – EDN WSSQVT.
  2. Kuchmenko, T. A. Use of piezosensors for determining the composition of the equilibrium gas phase of aqueous protein solutions / T. A. Kuchmenko, Y. A. Asanova // Russian Journal of Applied Chemistry. - 2009. - Vol. 82. - No 7. - P. 1188-1194. – DOI 10.1134/S1070427209070064. – EDN MWTUSN.
  3. Kuchmenko, T. A. Electronic nose based on nanoweights, expectation and reality / T. A. Kuchmenko // Pure and Applied Chemistry. - 2017. - Vol. 89. - No. 10. - P. 1587-1601. – DOI 10.1515/pac-2016-1108. – EDN XNMITO.
  4. A. Kuchmenko, T. Kuchmenko, R. U. U. Uhanov. - 2013. - Vol. 68. - No 4. - P. 368-375. – DOI 10.1134/S1061934813040059. – EDN RFFXUJ.

 For the studied polycomposite coatings, water vapor was also analyzed, however, the sensitivity of microbalancing (given in Table 1a) and the sorption kinetics of water vapor differ from the characteristics of the sorption of other VOCs, which allows them to be determined separately, including by the proposed parameters Aij.

Point 4. Author proposed that polycomposite coating was used to detect VOC and carried out a control experiment with a single coating, but did not specifically point out the advantages of polycomposite coating over a single coating.

Response 4: The advantage of using polycomposite coatings is the use of a smaller number of measuring elements in the electronic nose, the possibility of expanding the information obtained when they are used instead of standard single coating in already used devices. The possibility of increasing the sensitivity of microweighing when used in miniature devices. Briefly, the main advantages of polycomposite coatings are presented in Table No. 16 in the Conclusion section.

Point 5. The article mentioned collecting the exhaled gas of animals for sample analysis. Please explain how to selectively detect the required VOC among the many exhaled gas molecules. Please address the selectivity of the sensors.

Response 5: In this case, the article deals with the selective detection of vapors of substances in the gas phase above the exhaled breath condensate (EBC) at the level of 1 ppm. Earlier, we substantiated an approach to the identification of compounds by sensor signals using the Аij parameters, which is based on the fact that the ratio of sensor signals in a certain range of VOC concentrations can be analogous to the ratio of sensor sensitivity to VOC vapor and is determined by the nature of the detected compound. In this case, the choice of parameters as identification is determined by a number of requirements that they must meet. It was shown that only the closest homologues of a compound can have the same values of the identification parameter Аij. This approach and its application to the analysis of the gas phase over objects is described in articles mentioned in manuscript [26, 43-44, 64]. This article provides formulas for calculating analogues of the Aij parameter, only from the responses of a sensor with a polycomposite coating, which makes it possible to identify VOCs in gas mixtures of complex composition (in the article it is shown on the example of four-component mixtures), such as volatile compounds in the gas phase over EBC samples.

Point 6. Please explain the detection mechanism of the sensor.

Response 6: The mechanism of detection of volatile compounds by polycomposite coatings is as follows:

  1. The features of sorption of volatile compounds on individual coatings (one film per sensor) are studied: the time to reach the maximum signal, mechanism of sorption - surface (adsorption mechanism) or accumulation (absorption mechanism).
  2. A polycomposite coating is formed from the studied sorbents. Since the substance reacts simultaneously with all sensors in the detection cell, the specificity of the detection of volatile compounds is achieved through an processing algorithm taking into account the characteristics of the sorption of substances, i.e. the polycomposite coating does not specifically interact with the substance under study, however, after a detailed study of the mechanism of sorption on individual coatings, this information can be used to assess the presence of VOCs in gas mixtures or gas phases over real objects.

Our hypothesis is that when applying several sorbents to one transducer and reading its responses during the sorption of substances, the features of the sorption kinetics of this substance are preserved and they can be distinguished from the chronofrequencyogram by reading the sensor signals at certain moments of sorption.

Point 7. Some recent works related to VOC sensors are missing in the references. Coordination Chemistry Reviews 452 (2022) 214280; Sensors and Actuators A: Physical 2022, 347,113933. And it would be better to comment on the main advantages/disadvantages of the different sensor types.

Response 7: Thanks for the links to recent work by colleagues, we have expanded the introduction to the manuscripts. The main advantages/disadvantages of various types of sensors are shown in Table 1.

Reviewer 3 Report

This manuscript discussed a polycomposite coatings sensor for VOC vapor sorption.

Suggestions:

1, Figure 1 is missing. Would you please insert the figure?

2, Would you please list the legends in Figure 2?

3, Would you please explain what does the subscript mean in table 6?

Author Response

The authors are grateful for the study of the manuscript and valuable comments. Regarding the questions the following explanations are made:

Point 1. Figure 1 is missing. Would you please insert the figure?

Response 1: The manuscript has been corrected, figure 1 has been included in the text.

Point 2. Would you please list the legends in Figure 2?

Response 2: in Figure 2, we replaced the numbers with the captions for the curves of the graph and corrected the legend.

Point 3. Would you please explain what does the subscript mean in table 6?

Response 3: Subscripts in table 6 are corrected in accordance with the designations of sorbents adopted in the article.

Reviewer 4 Report

The paper shows applications of piezoelectric sensors in biomedicine and discuss its errors in detail. The manuscript is well written and presents very interesting results. To improve the quality of the manuscript, authors should answer the following questions:

 1. The authors of the article do not mention the reasons for choosing sensors for the analysis of exhaled breath condensate.

 2. In the text of the article there is no example of the responses of sensors with polycomposite coatings during the measurement of model mixtures or real objects.

 3. In the tables and the text of the article, for a better understanding of the laws of sorption, instead of the trivial names of compounds, it is better to give nomenclature names according to IUPAC (an example of an inconvenient name for the compound: Isobutyric acid, Isovaleric acid.)

Author Response

The authors are grateful for the study of the manuscript and valuable comments. Regarding the questions the following explanations are made:

Point 1. The authors of the article do not mention the reasons for choosing sensors for the analysis of exhaled breath condensate.

Response 1: The choice of sensors for the analysis of exhaled breath condensate was carried out on the basis of the experimental results according to the following criteria:

  1. Minimum system noise during measurement
  2. Operating stability
  3. High sensitivity to volatile disease markers
  4. Possibility of identification of compounds in gas mixtures.

This explanation was also added to the materials and methods section of the article.

Point 2. In the text of the article there is no example of the responses of sensors with polycomposite coatings during the measurement of model mixtures or real objects.

Response 2: Chronofrequency grams for sensors with polycomposite coatings for VOCs (for example, triethylamine and butyric acid) and exhaled air condensate samples have been added to Supplement materials, reference to them have also been added to the text of the article.

Point 3. In the tables and the text of the article, for a better understanding of the laws of sorption, instead of the trivial names of compounds, it is better to give nomenclature names according to IUPAC (an example of an inconvenient name for the compound: Isobutyric acid, Isovaleric acid.)

Response 3: the names of the compounds were corrected in accordance with IUPAC in the text of the article (isobutyric acid to 2-methylpropanoic acid, pentanoic acid to valeric, isovaleric acid to 3-methylbutanoic acid, butyric to butanoic acid).

Round 2

Reviewer 1 Report

The author has addressed all my concerns, I sugguest the manuscript to be published in Sensors.